# Adaptive Diffusion Terrain Generator for Autonomous Uneven Terrain Navigation

**Youwei Yu**[†]  **Junhong Xu**[†]  **Lantao Liu**
Indiana University, Bloomington
{youwyu, xu14, lantao}@iu.edu

**Abstract:** Model-free reinforcement learning has emerged as a powerful method for developing robust robot control policies capable of navigating through complex and unstructured terrains. The effectiveness of these methods hinges on two essential elements: (1) the use of massively parallel physics simulations to expedite policy training, and (2) an environment generator tasked with crafting sufficiently challenging yet attainable terrains to facilitate continuous policy improvement. Existing methods of environment generation often rely on heuristics constrained by a set of parameters, limiting the diversity and realism. In this work, we introduce the Adaptive Diffusion Terrain Generator (ADTG), a novel method that leverages Denoising Diffusion Probabilistic Models to dynamically expand existing training environments by adding more diverse and complex terrains adaptive to the current policy. ADTG guides the diffusion model's generation process through initial noise optimization, blending noise-corrupted terrains from existing training environments weighted by the policy's performance in each corresponding environment. By manipulating the noise corruption level, ADTG seamlessly transitions between generating similar terrains for policy fine-tuning and novel ones to expand training diversity. Our experiments show that the policy trained by ADTG outperforms both procedural generated and natural environments, along with popular navigation methods.

**Keywords:** Curriculum Reinforcement Learning, Guided Diffusion Model, Field Robots

## 1 Introduction

Autonomous navigation across uneven terrains necessitates the development of control policies that exhibit both robustness and smooth interactions within challenging environments [1, 2, 3]. In this work, we specifically target the training of a control policy that allows the mobile-wheeled robots to adeptly navigate through diverse uneven terrains, such as off-road environments characterized by varying elevations, irregular surfaces, and obstacles.

Recent advancements in reinforcement learning (RL) have shown promise in enhancing autonomous robot navigation on uneven terrains [4, 5, 6]. While an ideal scenario involves training an RL policy to operate seamlessly in all possible environments, the complexity of real-world scenarios makes it impractical to enumerate the entire spectrum of possibilities. Popular methods, including curriculum learning in simulation [7] and imitation learning using real-world collected data [8], encounter limitations in terms of training data diversity and the human efforts required. Without sufficient data and training, the application of learned policies to dissimilar scenarios becomes challenging, thereby hindering efforts to bridge the train-to-real gap. Additionally, existing solutions, such as scalar traversability classification for motion sampling [9, 10] and optimization methods [11, 12], may exhibit fragility due to sensor noise and complex characteristics of vehicle-terrain interactions.

---

[†]Equal contribution. Demonstrations at `https://adtg-sim-to-real.github.io`.

8th Conference on Robot Learning (CoRL 2024), Munich, Germany.

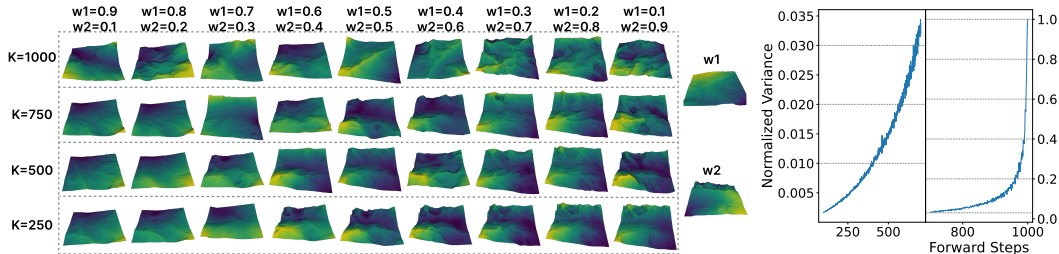

(a) Adjusting Realistic Terrain Diversity and Difficulty with ADTG     (b) Diffusion Variance

Figure 1: (a) Each row shows denoised terrains at different forward steps $K$, and each column blends terrains of varying difficulty using weighting factor $w$. Decreasing $w_1$ (easier terrain) and increasing $w_2$ raises difficulty. As $K$ increases, terrains show more novelty while maintaining difficulty. (b) Variance of a typical denoising diffusion process with a zoom-in view (left).

To tackle this challenge, we propose the Adaptive Diffusion Terrain Generator (ADTG), an environment generator designed to co-evolve with the policy, producing new environments that effectively push the boundaries of the policy's capabilities. Starting with an initial environment dataset, which may be from existing elevation data or terrains generated by generative models, ADTG is capable of expanding it into new and diverse environments. The significant contributions include:

1. **Adjustable Generation Difficulty:** ADTG dynamically modulates the complexity of generated terrains by optimizing the initial noise (latent variable) of the diffusion model. It blends noise-corrupted terrains from the training environments, guided by weights derived from the current policy's performance. As a result, the reverse diffusion process, starting at the optimized initial noise, can synthesize terrains that offer the right level of challenge tailored to the policy's current capabilities.

2. **Adjustable Generation Diversity:** By adjusting the initial noise level before executing the DDPM reverse process, ADTG effectively varies between generating challenging terrains and introducing new terrain geometries. This capability is tailored according to the diversity present in the existing training dataset, enriching training environments as needed throughout the training process. Such diversity is crucial to ensure the trained policy to adapt and perform well in a range of previously unseen scenarios.

We systematically validate the proposed ADTG framework by comparing it with established environment generation methods [13, 5] for training navigation policies on uneven terrains. Our experimental results indicate that ADTG offers enhanced generalization capabilities and faster convergence. Building on this core algorithm, we integrate ADTG with teacher-student distillation [14] and domain randomization [15] in physics and perception. We evaluate the deployment policy with zero-shot transfer to simulation and real-world experiments. The results reveal our framework's superiority over competing methods [16, 10, 4, 17] in key performance metrics.

## 2 Related Work

**Offroad Navigation.** Off-road navigation requires planners to handle more than simple planar motions. Simulating full terra-dynamics for complex, deformable surfaces like sand, mud, and snow is computationally intensive. Consequently, most model-based planners use simplified kinematics models for planning over uneven terrains [3, 18, 19, 20, 21] and incorporate semantic cost maps to evaluate traversability not accounted in the simplified model [2, 22]. Imitation learning (IL) methods [8, 23, 24] bypass terrain modeling by learning from expert demonstrations but require labor-intensive data collection. On the other hand, model-free RL does not require expert data and has shown impressive results enabling wheeled [7, 25, 4, 26] and legged robots [13, 5, 27, 28] traversing uneven terrains by training policies over diverse terrain geometries. However, the challenge is to generate realistic environments to bridge the sim-to-real gap. The commonly-used procedural generation methods [5, 13] are limited by parameterization and may not accurately reflect real-world terrain geometries. Our work addresses this by guiding a diffusion model trained on natural terrains to generate suitable terrain surfaces for training RL policies.

**Automatic Curriculum Learning and Environment Generation.** Our method is a form of automatic curriculum learning [29, 30], where it constructs increasingly challenging environments to train RL policies. While one primary goal of curriculum learning in RL is to expedite training efficiency [31, 32, 33], recent work shows that such automatic curriculum can be a by-product of unsupervised environment design (UED) [34, 35, 36, 37, 38]. It aims to co-evolve the policy and an environment generator during training to achieve zero-shot transfer during deployment. Unlike prior works in UED, the environments generated by our method are grounded in realistic environment distribution learned by a diffusion model and guided by policy performance. Recently, a concurrent work proposes Grounded Curriculum Learning [39]. It uses a variational auto-encoder (VAE) to learn realistic tasks and co-evolve a parameterized teacher policy to control VAE-generated tasks using UED-style training. In contrast, our work uses a sampling-based optimization method to control the diffusion model's initial noise for guided generation.

**Controllable Generation with Diffusion Models.** Controllable generation aims to guide a pre-trained diffusion model to generate samples that are not only realistic but also satisfy specific criteria. A commonly used strategy is adding guided perturbations to modify the generation process of a pre-trained diffusion model using scores from the conditional diffusion [40, 41] or gradients of cost functions [42]. Another approach is to directly optimize the weights of a pre-trained diffusion model so that the generated samples optimize some objective function. By treating the diffusion generation process as a Markov Decision Process, model-free reinforcement learning has been used to fine-tune the weights of a pre-trained diffusion model [43, 44]. This approach can also be viewed as sampling from an un-normalized distribution, given a pre-trained diffusion model as a prior [45]. Our work is closely related to initial noise optimization techniques for guiding diffusion models [46, 47, 48]. Instead of refining the diffusion model directly, these methods focus on optimizing the initial noise input. By freezing the pre-trained diffusion model, we ensure that the generated samples remain consistent with the original data distribution. In contrast to existing approaches focusing on content generation, our work integrates reinforcement learning (RL) with guided diffusion to train generalizable robotic policies.

## 3 Preliminaries

### 3.1 Problem Formulation

We represent the terrain using a grid-based *elevation* map, denoted as $e \in \mathbb{R}^{W \times H}$, where $W$ and $H$ represent the width and height, respectively. This terrain representation is widely adopted in motion planning across uneven surfaces. Similar to most works in training RL policies for rough terrain navigation [13, 5], we use existing high-performance physics simulators [49] to model the state transitions of the robot moving on uneven terrains $s_{t+1} \sim p(s_{t+1}|s_t, a_t, e)$. Here, $s \in \mathcal{S}$ and $a \in \mathcal{A}$ represent the robot's state and action, and each realization of the elevation (i.e., terrain) $e$ specifies a unique environment. An optimal policy $\pi(a|s, e; \theta)$ can be found by maximizing the expected cumulative discounted reward. Formally,

$$\theta^* = \arg\max_{\theta} \mathbb{E}_{\substack{a_t \sim \pi(a_t|s_t, e), s_0 \sim p(s_0), \\ e \sim p(e), s_{t+1} \sim p(s_{t+1}|s_t, a_t, e)}} \left[ \sum_{t=0}^{T} \gamma^t R(s_t, a_t) \right], \tag{1}$$

where $p(s_0)$ is the initial state distribution and $p(e)$ denotes the distribution over the environments. Due to the elevation $e$ imposing constraints on the robot's movement, the policy optimized through Eq. (1) is inherently capable of avoiding hazards on elevated terrains. We aim to dynamically evolve the environment distribution $p(e)$ based on the policy's performance, ensuring training efficiency and generating realistic terrain elevations. While constructing a realistic state transition $p(s_{t+1}|s_t, a_t, e)$ is also important for reducing the sim-to-real gap, we leave it to our future work.

### 3.2 Adaptive Curriculum Learning for Terrain-Aware Policy Optimization

A theoretically correct but impractical solution to Eq.(1) is to train on all possible terrains $\Lambda = (e^1, ..., e^N)$, with $p(e)$ as a uniform distribution over $\Lambda$. However, the vast variability of terrain

geometries makes this unfeasible. Even if possible, it might produce excessively challenging or overly simple terrains, risking the learned policy to have poor performance [50]. Adaptive Curriculum Reinforcement Learning (ACRL) addresses these issues by dynamically updating the training dataset [51]. ACRL generates and selects environments that yield the most policy improvement. In our work, designing an effective environment generator is crucial. It should (1) generate realistic environments matching real-world distributions and (2) adequately challenge the current policy. Common approaches include using adjustable pre-defined terrain types [13], which offers control but may lack diversity, and generative models [52], which excel in realism but may struggle with precise policy-tailored challenges. In the following, we detail the proposed ADTG, which balances realism and policy-tailored terrain generation.

## 4 Adaptive Diffusion Terrain Generator

This section introduces the Adaptive Diffusion Terrain Generator (ADTG), a novel ACRL generator that manipulates the DDPM process based on current policy performance and dataset diversity. We begin by interpolating between "easy" and "difficult" terrains in the DDPM latent space to generate terrains that optimize policy training. Next, we modulate the initial noise input based on the training dataset's variance to enrich terrain diversity, fostering broader experiences and improving the policy's generalization across unseen terrains. We use $e$, $e_0$, and $e_k$ to denote the environment in the training dataset, the generated terrain through DDPM, and the DDPM's latent variable at timestep $k$, respectively. All three variables are the same size $\mathbb{R}^{W \times H}$. Since in DDPM, noises and latent variables are the same [53], we use them interchangeably.

### 4.1 Performance-Guided Generation via DDPM

**Latent Variable Synthesis for Controllable Generation.** Once trained, DDPM can control sample generation by manipulating intermediate latent variables. In our context, the goal is to steer the generated terrain surface to maximize policy improvement after being trained on it. While there are numerous methods to guide the diffusion model [40, 44], we choose to optimize the starting noise to control the final target [46]. This approach is both simple and effective, as it eliminates the need for perturbations across all reverse diffusion steps, as required in classifier-free guidance [40], or fine-tuning of diffusion models [44]. Nevertheless, it still enhances the probability of sampling informative terrains tailored to the current policy.

Consider a subset of terrain elevations $\bar{\Lambda} = (e^1, e^2, \ldots, e^n)$ from the dataset $\Lambda$. To find an initial noise that generates a terrain maximizing the policy improvement, we first generate intermediate latent variables (noises) for each training environment in $\bar{\Lambda}$ at a forward diffusion time step $k$, $e_k^i \sim q(e_k^i | e^i, k)$ for $i = 1, \ldots, n$. Assume that we have a weighting function $w(e, \pi)$ that evaluates the performance improvement after training on each terrain map $e^i$. We propose to find the optimized initial noise as a weighted interpolation of these latents, where the contribution of each latent $e_k^i$ is given by the policy improvement in the original terrain environment $w(e^i, \pi)$.

$$e'_k = \left[ \Sigma_{i=1}^n w(e^i, \pi) e_k^i \right] / \left[ \Sigma_{m=1}^n w(e^m, \pi) \right]. \tag{2}$$

The fused latent variable $e'_k$ is then processed through reverse diffusion, starting at time $k$ to synthesize a new terrain $e'_0$. The resulting terrain blends the high-level characteristics captured by the latent features of original terrains, proportionally influenced by their weights. We illustrate this blending effect in Fig. 1. By controlling weights $w_i$, we can steer the difficulty of the synthesized terrain.

**Weighting Function.** Policy training requires dynamic weight assignment based on current policy performance. We define the following weighting function that penalizes terrains that are too easy or too difficult for the policy:

$$w(e, \pi) = \exp\{r(e, \pi)\}, \quad r(e, \pi) = -(\mathfrak{s}(e, \pi) - \bar{\mathfrak{s}})^2 / \sigma^2. \tag{3}$$

It penalizes the deviation of *terrain difficulty*, $\mathfrak{s}(e, \pi)$, experienced by the policy $\pi$ from a desired difficulty level $\bar{\mathfrak{s}}$. This desired level indicates a terrain difficulty that promotes the most significant

improvement in the policy. The temperature parameter $\sigma$ controls the sensitivity of the weighting function to deviations from this desired difficulty level. We use the navigation success rate [54] to represent $\mathfrak{s}(\cdot, \cdot)$. While alternatives like TD-error [55] or regret [56] exist, this metric has proven to be an effective and computationally efficient indicator for quantifying an environment's potential to enhance policy performance in navigation and locomotion tasks [5, 13]. In Appendix A, we show that the optimized noise in Eq. (2) and the corresponding weighting function in Eq. (3) can be derived from formulating the noise optimization problem using Control as Inference [57, 58] and solving it through Importance Sampling. We denote the procedure of optimizing the noise $e'_k$ using Eq. 2 and generating the final optimized environment by reverse diffusion starting at $e'_k$ as $e' = \mathtt{Synthesize}(\bar{\Lambda}, \pi, k)$, where $k$ is the starting time step of the reverse process. As discussed in the next section, a large $k$ is crucial to maintaining diversity.

## 4.2 Diversifying Training Dataset via Modulating Initial Noise

The preceding section describes how policy performance guides DDPM in generating terrains that challenge the current policy's capabilities. As training progresses, the pool of challenging terrains diminishes, leading to a point where each terrain no longer provides significant improvement for the policy. Simply fusing these less challenging terrains does not create more complex scenarios. Without enhancing terrain diversity, the potential for policy improvement plateaus. To overcome this, it is essential to shift the focus of terrain generation towards increasing diversity. DDPM's reverse process generally starts from a pre-defined forward step, where the latent variable is usually pure Gaussian noise. However, it can also start from any forward step $K$ with sampled noise as $e_K \sim q(e_K|e_0)$ [59]. Fig. 1 shows the variance of generated terrains decreases with fewer forward steps and vice versa. To enrich our training dataset's diversity, we propose the following:

1. **Variability Assessment**: Compute the dataset's variability $\Lambda_{var}$ by analyzing the variance of the first few principal components from a Principal Component Analysis (PCA) on each elevation map. This serves as an efficient proxy for variability.

2. **Forward Step Selection**: The forward step $k \propto \Lambda_{var}^{-1}$ is inversely proportional to the variance. We use a linear scheduler: $k = K(1 - \Lambda_{var})$, with $K$ the maximum forward step and $\Lambda_{var}$ normalized to $0 \sim 1$. This inverse relationship ensures greater diversity in generated terrains.

3. **Terrain Generation**: Using the selected forward step $k$, apply our proposed $\mathtt{Synthesize}$ to generate new terrains, thus expanding variability for training environments.

## 4.3 ACRL with ADTG

We present the final method pseudo-coded in Alg. 1 using the proposed ADTG for training a privileged policy. The algorithm iterates over policy optimization and guided terrain generation, co-evolving the policy and terrain dataset until convergence. The algorithm starts by selecting a training environment that provides the best training signal for the current policy, which can be done in various ways [50]. For example, one can compute scores for terrains based on the weighting function in Eq. (3) and choose the

**Algorithm 1** ACRL with Adaptive Diffusion Terrain Generator

**Input:** Pretrained DDPM $\epsilon(\cdot, \cdot; \phi)$, an initial terrain dataset $\Lambda$
**Output:** The optimized privileged policy $\pi^*$
**Initialize:** The privileged policy $\pi$
1: **while** $\pi$ not converge **do**
2:     $e = \mathtt{Selector}(\Lambda, \pi)$    ▷ Env. Selection
3:     $\pi \leftarrow \mathtt{Optim}(\pi, e)$         ▷ Policy Update
4:     $k = K(1 - \Lambda_{var})$           ▷ Sec. 4.2
5:     $e'_0 = \mathtt{Synthesize}(\Lambda, \pi, k)$  ▷ Sec. 4.1
6:     $\Lambda \leftarrow \Lambda \cup e'_0$         ▷ Update Dataset
7: **end while**

one with the maximum weight. Instead of choosing deterministically, we sample the terrain based on their corresponding weights. The $\mathtt{Optim}$ step collects trajectories and performs one policy update in the selected terrain. After the update, we evolve the current dataset by generating a new one, as shown in lines 4 - 6 of Alg. 1. In practice, we run Alg. 1 in parallel across $N$ terrains, each with multiple robots. In parallel training, $\mathtt{Synthesize}$ begins by sampling $N \times n$ initial noises, where $N$ is the number of new terrains (equal to the number of parallel environments) and $n$ is the sample size in Eq. (2). It then optimizes over these noises to generate $N$ optimized noises. Finally, these optimized noises are passed to the DDPM to generate $N$ terrains. When the dataset grows large,

it sub-samples terrains from `Selector`'s complement, with success rates updated by the current policy. We validate effectiveness of `Selector` in Sec. 5.1 and `Synthesize` in Appendix A.3, which also explains its sub-sampling logic.

**ADTG with Policy Distillation.** While ADTG can train policies to generalize over terrain geometries, real-world deployments face challenges beyond geometry. This includes noisy, partial observations and varying physical properties. To address these, we distill a privileged policy, which observes the complete elevation map and noiseless states into a depth vision-based policy with noisy measurements. The privileged policy is trained using PPO [60], and the deployment policy is trained using DAgger [61], utilizing methods similar to [6]. To enhance generalization, we integrate physical and perception domain randomization [15] (more details in Appendix. C). The training loop integrating ADTG with teacher-student distillation is illustrated in Fig. 2 and the Appendix. With ADTG validated in Sec. 5.1, our whole system is demonstrated in the sim-to-deploy experiments.

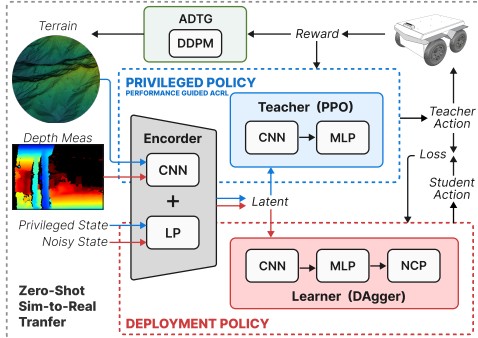

Figure 2: Framework with our Adaptive Diffusion Terrain Generation (ADTG) and Policy Distillation. Model-free RL trains privileged policy on ADTG-generated terrains. The privileged policy is then distilled into the deployment (Learner) policy using data aggregation. Iterative training and terrain generation through ADTG enhance the deployment policy's generalization.

## 5 Experiments

In this section, we start with the algorithmic evaluation to highlight the effectiveness of our ADTG. Then through sim-to-sim on two kinds of wheeled robot platforms and sim-to-real on one wheeled and one quadruped robot, we study the performance of our ADTG policy against ablations and competing methods.

We train in IsaacGym [49] and parallel 100 uneven terrains, each with 100 robots. Among the elevation dataset [62] as detailed in Appendix C.1, we select 3000 for DDPM training (E-3K), 100 for algorithmic evaluation (E-1H), and 30 for sim-to-sim experiments (E-30). Simulations run on an Intel i9-14900KF CPU and NVIDIA RTX 4090 GPU. Real-world tests use an NVIDIA Jetson Orin.

### 5.1 Algorithmic Performance Evaluation

This section evaluates whether the environment curriculum generated by ADTG enhances the generalization capability of the trained privileged policy across unfamiliar terrain geometries, on the wheeled ClearPath Jackal robot. We compare with the following baselines. Procedural Generation Curriculum (**PGC**), a commonly used method, uses heuristically designed terrain parameters [13]. Our implemented

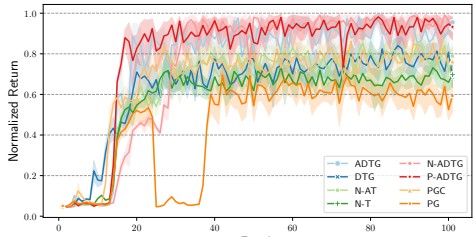

Figure 3: The comparison of the normalized return among our proposed ADTG, the baselines, and ablation methods.

PGC follows ADTG, adapting the terrain via the score function Eq. (3) and dynamically updating the dataset. First, to ablate our Adaptive curriculum, Diffusion Terrain Generator (A**DTG**) generates terrain without curriculum. Procedural Generation (**PG**C) randomly samples parameters. To ablate our Diffusion Generator, Natural Adaptive Terrain (**N-A**D**T**G) selects terrains directly from E-3K. To ablate both, Natural Terrain (**N-**AD**T**G) randomly samples from E-3K without curriculum. `Mono` font means the ablated parts. Second, to test ADTG's robustness to initialization other than diffusion, **N-ADTG** and **P-ADTG** start with datasets sub-sampled from E-3K and procedural generations.

All methods use the same training and evaluation setup. After each training epoch, policies are tested in *held-out* evaluation environments with 200 000 start-goal pairs. Fig. 3 shows the normalized RL return[1]. ADTG outperforms PGC and N-AT within 40 epochs, exceeding the return by over 0.1,

---

[1]It is calculated by the actual return divided by value function's upper bound based on our reward function.

| | Suc. Rate (%) | | | Traj. Ratio | | | Orien. Vib. ($\frac{rad}{s}$) | | | Orien. Jerk ($\frac{rad}{s^3}$) | | | Pos. Jerk ($\frac{m}{s^3}$) | | |
|---|---|---|---|---|---|---|---|---|---|---|---|---|---|---|---|
| | S | R | D | S | R | D | S | R | D | S | R | D | S | R | D |
| Falco | 26 | 47 | 22 | 2.76 | 1.24 | 1.18 | 0.71 | 0.15 | 0.17 | 275.6 | 165 | 143.2 | 48 | 42.1 | 20.3 |
| MPPI | 48 | 71 | 34 | **1.21** | 1.23 | **1.04** | 0.75 | 0.18 | 0.18 | 228.7 | 161.7 | 81 | 40.7 | 34.2 | 23.6 |
| TERP | 33 | 68 | 25 | 1.62 | **1.16** | **1.1** | 0.77 | 0.13 | 0.13 | **210.1** | 137.6 | 112.2 | **37.1** | **20.2** | **17.5** |
| POVN | 17 | 28 | 24 | **1.23** | 1.23 | 1.28 | **0.68** | 0.18 | 0.14 | 241 | 120.2 | 127.5 | 43.7 | 38.6 | 21.3 |
| N-AT | **67** | x | x | 1.24 | x | x | 1.08 | x | x | 323.4 | x | x | 57.6 | x | x |
| PGC | 43 | x | x | 1.92 | x | x | 0.97 | x | x | 236.1 | x | x | 42 | x | x |
| Ours | **87** | **80** | **45** | 1.52 | **1.25** | 1.32 | **0.65** | **0.11** | **0.13** | **193.5** | **113.3** | **73.4** | **35** | **20** | 18.1 |

Table 1: Statistical results for S(imulation), R(eal-world) and D(une) comparing our method with SOTA works. S: 30 environments; R: grass, forest, arid, rust, mud, and gravel; D: dune hard, tough, and expert. **Green** and **Bold** indicate the best and second-best results. x means poor performance.

which translates to more than 20 000 successes in our recorded data. As shown in N-ADTG and P-ADTG, regardless of initial terrains, ADTG consistently generates effective terrains. DTG, N-T, and PG results validate our curriculum `Selector`, with PG's sharp curve changes due to overly difficult terrains. N-AT lacks dataset evolution, and PGC lacks efficient terrain parameter control. In summary, ADTG excels at adapting environment difficulty based on evolving policy performance.

## 5.2 Zero-shot Sim-to-Sim and Sim-to-Real Experiments

This section evaluates the zero-shot transfer capability of sim-to-deploy environments. Metrics include the success rate, trajectory ratio, orientation vibration $|\omega|$, orientation jerk $|\frac{\partial^2 \omega}{\partial t^2}|$, and position jerk $|\frac{\partial a}{\partial t}|$, where $\omega$ and $a$ denote angular velocity and linear acceleration. These motion stability indicators are crucial in mitigating sudden pose changes and enhancing overall safety. The trajectory ratio is the successful path length relative to straight-line distance and indicates navigator efficiency. All baselines use the elevation map [63] with depth camera and identify terrains as obstacles if the slope estimated from the elevation map exceeds $20°$. For orientation costs, we obtain the robot's roll and pitch by projecting its base to the elevation map.

**Baselines. Falco** [16], a classic motion primitives planner, and **MPPI** [10, 64], a sampling-based model predictive controller, are recognized for the success rate and efficiency. They use the point-cloud and elevation map to weigh collision risk and orientation penalty. **TERP** [4], an RL policy trained in simulation, conditions on the elevation map, rewarding motion stability and penalizing steep slopes. **POVN**av [17] performs Pareto-optimal navigation by identifying sub-goals in segmented images [65], excelling in unstructured outdoor environments.

**Simulation Experiment.** We simulate wheeled robots, ClearPath Jackal and Husky, in ROS Gazebo on 30 diverse environments (E-30), equipped with a RealSense D435i camera. We add Gaussian noises to the robot state, depth measurement, and vehicle control to introduce uncertainty. 1000 start and goal pairs are sampled for each environment. We do not include ablations other than N-AT because of poor algorithmic performance. As Jackal's results shown in Table 1, our method outperforms the baselines. Appendix C.2 provides statistical results for Husky. While all methods show improved performance due to the Husky's better navigability on uneven terrains, our method consistently outperformed baseline methods. The depth measurement noise poses a substantial challenge in accurately modeling obstacles and complex terrains. Falco and MPPI often cause the robot to get stuck or topple over, and TERP often predicts erratic waypoints that either violate safety on elevation map or are overly conservative. Learning-based TERP and POVN lack generalizability, with their performance varying across different environments. This issue is mirrored in N-AT and PGC, highlighting the success of adaptive curriculum and realistic terrain generation properties of ADTG.

**Real-world Experiment.** The Jackal robot is equipped with a Velodyne-16 LiDAR, a RealSense D435i camera, and a 3DM-GX5-25 IMU. We test in 9 diverse representative environments, `grass`, `forest`, `arid`, `rust`, `mud`, `gravel`, `dune-hard`, `dune-tough`, and `dune-expert`, displayed in Fig. 4. For each environment, we sample 36 start-goal pairs. Note that the results of N-AT and PGC policies are not presented as they achieved limited success in real-world experiments. Table 1 provides an overview of the performance metrics, with success rates across all methods de-

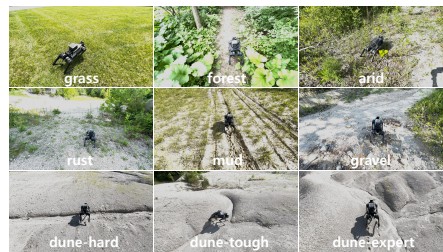
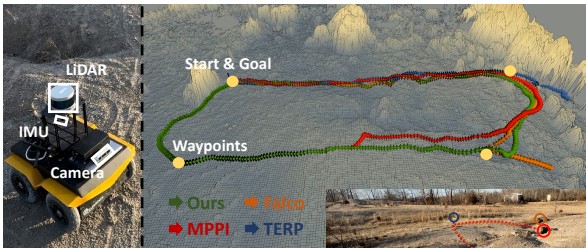

(a) Experimental Environments       (b) Robot and Dune-Hard Example Demonstrations

Figure 4: The left panel shows nine challenging environments, the middle our platform, and the right Dune-hard, where our method outperformed others in navigating ravines.

tailed in the Appendix C.3. In `forest`, POVNav struggled with image segmentation of complex ground elements like foliage and underbrush. Other baselines encountered difficulties due to imprecise depth data affecting terrain modeling. All baselines risked getting stuck in muddy, sandy, and rocky areas. In contrast, our method autonomously engaged in arc movements, avoiding immobilization in slippery environments. In `dunes`, characterized by steep elevation changes, failures of other methods were due to incorrect obstacle detection or severe slope changes causing overturn.

In addition, we ablated **Physics** and **Perception** domain randomization to study their contributions. Removing either reduced success rates, though both ablations outperformed baselines due to ADTG. The perception domain suffered from camera depth issues tied to exposure. Without the physics domain, the robot showed improved orientation smoothness but struggled to make reactive behaviors, since the learned arc movements in slippery areas benefited from this domain.

**Real-World Quadruped Locomotion.** To validate ADTG's generalization across different embodiments, we benchmark against PGC and the built-in MPC controller on the Unitree Go1 quadruped in the same environments as Jackal. Both ADTG and PGC were trained using the Parkour [66] "walk" policy. Similar to the hiking experiment [5, 13], we conducted a continuous $1.2\,\mathrm{km}$ loop, following MPC's footprints for fairness. Failures, defined as rollovers, occurred when detours around challenging areas were not allowed. MPC had 6 rollovers 3 from high-speed commands, 3 on challenging gravel and dune terrains). ADTG policy exhibited natural postures with 2 failures due to high-speed commands. PGC policy showed robust prostrate postures but 11 failures due to sudden movements (jumps) on all terrains except grass. The results highlight ADTG's advantages in learning robust behavior. See the appendix for the full Jackal ablation analysis and quadruped experiment details.

## 6 Conclusion, Limitations and Future Directions

We propose an Adaptive Diffusion Terrain Generator (ADTG) to create realistic and diverse terrains based on evolving policy performance, enhancing RL policy's generalization and learning efficiency. To guide the diffusion model generation process, we propose optimizing the initial noises based on the potential improvements of the policy after being trained on the environment denoised from this initial noise. Algorithmic performance shows ADTG's superiority in generating challenging but suitable environments over established methods such as commonly used procedural generation curriculum. Combined with domain randomization in a teacher-student framework, it trains a robust deployment policy for zero-shot transfer to new, unseen terrains. Extensive sim-to-deploy tests with wheeled and quadruped robots validate our approach against SOTA planning methods.

**Limitations and Future Directions:** A key limitation of ADTG is that it evolves only the environment distribution, relying on physics simulators for state transitions, limiting deployment in the complex real world. While domain randomization helps, it's not a full solution. Future work will integrate environment distribution and physics to bridge the sim-to-real gap. Additionally, ADTG's environment scale is suited for local planning, but larger environments are needed for long-horizon tasks. We plan to explore hierarchical diffusion models for generating multi-layered environments.

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

# Appendix

## A  Initial Noise Optimization by Control as Inference

In this section, we justify our algorithm design through the lens of control-as-inference [1, 2] to optimize the initial noise of diffusion models.

### A.1  Denoising Diffusion Probabilistic Models Preliminary

Denoising Diffusion Probabilistic Models (DDPMs) [5] are generative models that generate original data from Gaussian noise through a series of forward and reverse steps. In the forward process, Gaussian noise is gradually added to the original terrain sample $e_0$ until a final step $K$, with the terrain at each step $k$ distributed as $q\left(e_k|e_{k-1}\right) = \mathcal{N}\left(e_k|\sqrt{1 - \beta_k}e_{k-1}, \beta_k I\right)$. We use cosine noise schedules [6] $(\beta_1, ..., \beta_K)$. The reverse process gradually reduces the noise from $e_K$ to generate the original data $e_0$ using a noise predictor $\epsilon(e_k, k; \phi)$. The training loss function for the noise predictor is $\mathcal{L}(\phi) = \text{MSE}(\epsilon_k, \epsilon(e_0 + \epsilon_k, k; \phi))$.

### A.2  DDPM Initial Noise Optimization

Let $p(e_{k-1}|e_k; \theta)$ denote single reverse diffusion step, with $\theta$ the diffusion model's parameters. Based on [3], we can formulate iterative denoising of the DDPM as a Markov Decision Process (MDP), where the state evolution (policy and transition function combined) is the reverse diffusion itself $p(s_{k+1}|s_k, a_k) = p(e_{k-1}|e_k; \theta)$, and the reward function is 0 unless at the initial step $k = 0$:

$$R(e_k, \pi) = \begin{cases} r(e_0, \pi) & \text{if } k = 0 \\ 0 & \text{otherwise} \end{cases} \tag{4}$$

In our work, $r(e_0, \pi)$ evaluates the improvements of the current policy $\pi$ if trained on the generated environment $e_0$. The return of the noise $e_k$ starting at timestep $k$ as

$$\mathcal{J}(e_k) = \mathbb{E}\left[r(e_0, \pi)|e_{t-1} \sim p(e_{t-1}|e_t; \theta)\right], \tag{5}$$

where $t \in [1, k]$. Our goal is to find the noise that maximizes the Eq. (5), $e_k^* = \arg\max_{e_k} \mathcal{J}(e_k)$. Gradient-based methods using differentiable rewards [4] are challenging to apply in our setup, as computing the gradient of policy improvement with respect to the initial noise is impractical. This is because evaluating policy improvement requires simulating the policy on each generated terrain to compute success rates. Instead, we frame this as a control-as-inference problem, using approximate sampling to optimize the initial noise for Eq. (5). Following the KL control theory derivation [1], we aim to find a distribution of the starting noise $\hat{q}(e_k)$ that optimizes Eq. (5) while remaining close to a reference noise distribution $q(e_k)$. This is achieved by adding an extra KL cost to the return in Eq. (5) and taking expectation with respect to the random initial noise

$$\hat{\mathcal{J}}(e_k) = \mathbb{E}\left[\mathcal{J}(e_k) - \log\frac{\hat{q}(e_k)}{q(e_k)}\bigg|e_k \sim \hat{q}(e_k)\right] \tag{6}$$

$$= \int \hat{q}(e_k)\left(\mathcal{J}(e_k) - \log\frac{\hat{q}(e_k)}{q(e_k)}\right)de_k$$

$$= \int \hat{q}(e_k)\left(\log\left(\exp\{\mathcal{J}(e_k)\}\right) - \log\frac{\hat{q}(e_k)}{q(e_k)}\right)de_k$$

$$= -\int \hat{q}(e_k)\log\frac{\hat{q}(e_k)}{q(e_k)\exp\{\mathcal{J}(e_k)\}}de_k$$

$$= -\text{KL}\left(\hat{q}(e_k)\|\psi(e_k)\right),$$

where $\psi(e_k) \propto q(e_k)\exp\mathcal{J}(e_k)$ is an unnormalized distribution with high density at the high return region. The Kullback-Leibler (KL) divergence is non-negative, implying that the optimal return is achieved when $\hat{q} = \psi$. Consequently, sampling from $\psi$ is equivalent to sampling from the optimal noise distribution. While there are multiple methods to sample from $\psi$, we employ importance

sampling to compute the expected value of the noise. This approach is chosen due to its simple computational structure within this formulation. The importance sampling can then be expressed as

$$\mathbb{E}\left[e_k | e_k \sim \psi(e_k)\right] = \frac{1}{Z} \int e_k q(e_k) \exp\left\{\mathcal{J}(e_k)\right\} de_k \tag{7}$$

$$= \frac{1}{Z} \mathbb{E}\left[\exp\left\{\mathcal{J}(e_k)\right\} \mid e_k \sim q(e_k)\right]$$

$$\approx \frac{1}{Z} \frac{1}{N} \sum_{i=1}^{N} \exp\left\{\mathcal{J}(e_k)\right\} e_k,$$

where the samples $e_k$ are drawn from the reference distribution. The normalization constant can be estimated similarly

$$Z = \int q(e_k) \exp\left\{\mathcal{J}(e_k)\right\} de_k \tag{8}$$

$$= \mathbb{E}\left[\exp\left\{\mathcal{J}(e_k)\right\} | e_k \sim q(e_k)\right]$$

$$\approx \frac{1}{N} \sum_{i=1}^{N} \exp\left\{\mathcal{J}(e_k)\right\}.$$

Combining Eq.(7) and Eq.(8), we obtain the expected optimal noise

$$\mathbb{E}\left[e_k | e_k \sim \psi(e_k)\right] \approx \frac{\sum_{i=1}^{N} \exp\left\{\mathcal{J}(e_k)\right\} e_k}{\sum_{i=1}^{N} \exp\left\{\mathcal{J}(e_k)\right\}}. \tag{9}$$

We define the reference distribution as the marginal distribution of the forward diffusion process $q(e_k) = \int q(e_k|e)p(e)de$, where $p(e)$ is the distribution over the current training environment dataset. Sampling from $q(e_k)$ involves drawing from $p(e)$ and then from $q(e_k|e)$, which is exactly the process described in Section 4.1. This process is efficient because the forward diffusion $q(e_k|e)$ is easy to sample, and we can readily draw samples from $p(e)$ using the available environment dataset. The steps in our algorithm directly correspond to the computation of the final expected noise as expressed in Eq. (7). Table 2 illustrates the mapping between these algorithmic steps and the computational steps of computing Eq. (7).

| Control as Inference | ADTG |
| --- | --- |
| Action/State space | Space of initial noise |
| Optimal policy | Optimal initial noise distribution |
| Optimal action | Optimal initial noise $e_k'$ |
| Reward | Negative deviation from the desired success rate $-(\mathfrak{s}(e, \pi) - \bar{\mathfrak{s}})^2/\sigma^2$ |
| Sample from proposal $e_k \sim q(e_k)$ | Sample from training dataset $e \sim p(e)$ (Appendix A.3) Sample from forward diffusion process $e_k \sim q(e_k|e)$ |
| Importance weight | Weighting function $\exp r(e, \pi)$ (Eq. (3)) |
| Importance sampling solver (Eq. (7)) | Weighted interpolation (Eq. (2)) |

Table 2: Correspondence of algorithmic steps in ADTG with control as inference using importance sampling.

## A.3 Environment Difficulty Manipulated by Diffusion Synthesis

The core assumption underlying our method is that the success rates of the generated terrains by ADTG are consistent with the weighted combination of the success rates of selected terrains. This appendix provides empirical evidence to support this assumption. During training, we sample terrains with the success rate outside the $0.6$ to $0.85$ range for `Synthesize`. As our dataset expands rapidly, we uniformly sub-sample up to 1000 terrains as the current (sub-)dataset. If the synthesized terrain's predicted success rate falls outside the $0.6$ to $0.85$ range, we adjust by adding suitable samples while ensuring the total number under 16. We choose 16 based on the GPU memory

limitation. We first set aside 100 terrains as dataset $\mathcal{D}$ and assess their success rates using the privileged policy. For each synthesized terrain, `Selector` sub-samples $N$ terrains from $\mathcal{D}$, where $N$ follows the curriculum. For each synthesized terrain, we define the predicted success rate as $\mathfrak{s}' = (\sum_{i=1}^{n} w_i \mathfrak{s}_i)/(\sum_{i=1}^{n} w_i)$, where $w$ and $\mathfrak{s}$ denote `Synthesize` weight and success rate, respectively. The validation involves comparing the actual success rates evaluated by the privileged policy against the predicted success rates. Fig. 5 illustrates this comparison across 8 diffusion steps, with each step containing 60 synthesized samples. For clarity, each step's display is organized into two rows. The first row presents the predicted success rates, and the second row shows the actual success rates. The similarity in color between these two rows indicates the consistency of the success rates for the generated terrains. Our result shows that most of the samples closely align with the predicted success rates.

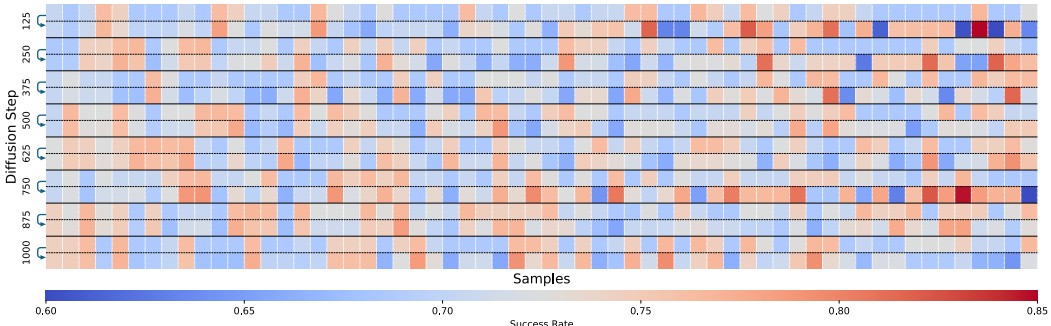

Figure 5: Synthesized terrains' success rates by diffusion model. For each diffusion step with 60 samples, the first row presents predicted success rates, and the second shows actual success rates. This shows effective terrain difficulty manipulation by our `Synthesize` function.

## B    System Design

### B.1    Privileged Policy

The privileged (teacher) policy is trained using Proximal Policy Optimization (PPO) [7] for goal-oriented navigation on uneven terrains, minimizing motion vibration and jerk. The design is as following:

**State, Observation, and Action.** The state $s_t^{\text{tr}} = [q_t, v_t, \omega_t, d_t, a_{t-1}^{\theta}]$ at timestamp $t$ is ground-truth motion information, including the orientation in quaternion $q_t \in \mathbb{R}^4$, linear velocity $v_t \in \mathbb{R}^3$, angular velocity $\omega_t \in \mathbb{R}^3$, relative goal distance $d_t \in \mathbb{R}^2$, and previous action $a_{t-1}^{\theta} \in \mathbb{R}^2$. The reference frame is anchored to the robot base. Normalization is applied to linear and angular velocities. We refrain from normalizing the relative goal distance, as this allows the policy to adapt to varying goal range scales. The observation has full access to the elevation map $o^{\text{tr}} \in \mathbb{R}^{128 \times 128}$. Each observation remains constant during an ACRL episode when ADTG does not evolve terrains. The action $a_t^{\text{tr}} = [\alpha_v v_t^x, \alpha_\omega \omega_t^z]$ applied to the robot represents proportional-derivative (PD) targets for forward linear and yaw angular velocities, where $a_t^\theta = [v_t^x, \omega_t^z]$ is the network output, and $[\alpha_v, \alpha_\omega]$ are coefficients for the vehicle drive system.

**Rewards.** The total reward is structured as a sum of the weighted components: Goal Proximity $c_1 \mathbb{1}_{\delta_g}(s_t)$ assesses the robot's closeness to the goal against thresholds $\delta_{\text{pos}}$ and $\delta_{\text{rot}}$, Orientation Regulation $c_2 \mathbb{1}_{\delta_e}(E(q_t))$ imposes penalties for exceeding safe roll and pitch angles, Movement Consistency $c_3 ||a_t - a_{t-1}||_2 + c_4 ||\dot{a}_t - \dot{a}_{t-1}||_2$ incentives consistent and gradual actions, Ground-contact and Safety $c_5 \dim(f = 0) + c_6 \dim(f > \delta_F)$ discourages situations where the contact force mean ground contact loss or a collision risk. The total reward is parameterized as follows:

$$
\begin{aligned}
r = &5.0 \times \mathbb{1}_{\delta_{\text{pos}} \leq 0.25}(s_t) + 0.001 \times \mathbb{1}_{\delta_{\text{rot}} \leq \pi/3}(s_t) \\
&- 0.01 \times \mathbb{1}_{\delta_e > \pi/12}(q_t) \\
&- 0.01 \times ||a_t - a_{t-1}||_2 - 0.0001 \times ||\dot{a}_t - \dot{a}_{t-1}||_2 \\
&- 0.01 \times \dim(f > 100) - 0.001 \times \dim(f \leq 0.00001).
\end{aligned}
\tag{10}
$$

**Start and Termination.** Episodes start with robots in random poses and assigned navigation targets. The initial state $s_0$ is sampled by position, velocity, and yaw, with other parameters managed by the physics engine. The state $z$ considers terrain geometry to avoid immediate failure. Initially, goals are within a circular sector, with a radius of $d_{\max} = [0.5, 3.0]$m and a central angle of $[-\pi/3, \pi/3]$rad. As the policy progresses to success rate of $60\%$, the radius and angle incrementally increase by 0.5m and $\pi/18$ until a semicircle with a radius of 6m. An episode ends upon reaching the goal or triggering safety or running length constraints. The maximum episode length is $d_{\max}/\Delta t/v_{\mathrm{avg}}$, with $v_{\mathrm{avg}} = 0.3$ m/s the minimum average velocity and $\Delta t = 0.005$ s the simulator timestep. Early termination occurs if the roll or pitch angle exceeds $\delta_{\mathrm{quat}} = \pi/9$, or if the robot collides with the environment, or if two or more wheels are off the ground, risking toppling or entrapment.

## B.2 Deployment Policy

The deployment (student) policy $\hat{\theta}$ is trained via Dataset Aggregation (DAgger) [8] to minimize the mean squared error to match the teacher's actions. At each timestamp $t$, the deployment policy observes $o_t = (\tilde{s}_t, I_t)$, where $\tilde{s}_t$ is the noisy state and $I_t$ is a depth image. The depth sensor captures calibrated depth images with $[640 \times 480]$ resolution and $87°$ horizontal field-of-view, which mirror our sim-to-deploy experimental settings. Due to partial observability, the policy considers past information to decide the next action $a_t \sim \hat{\pi}(a_t|\boldsymbol{a}_t, \mathbf{o}_t; \hat{\theta})$, where $\boldsymbol{a}_t$ and $\mathbf{o}_t$ are action and observation histories, $H$ is the maximum history length. To enhance generalization, we integrate physical domain randomization and perception domain randomization as following:

**Physics Domain Randomization.** An environment appears as geometry and is characterized by physics $e^p = \{e^{p_v}, e^{p_g}; e^{p_m}, e^{p_f}, e^{p_a}\} \in \mathbb{R}^{10}$, which is separated into environment properties and robot-environment interactions. $e^{p_v}$: Dynamic friction, static friction, and restitution coefficients, affecting slipperiness; $e^{p_g}$: Gravity; $e^{p_m}$: Mass, simulating extra burdens or flat tires; $e^{p_f}$: External forces; $e^{p_a}$: Discrepancies in actuator setpoints.

**Perception Domain Randomization.** Since velocity relies on the inertial measurement unit (IMU), which is typically noisy [14], we add Gaussian noise to simulate this: $\tilde{v} \sim \mathcal{N}(v, \sigma_v^2)$ for linear velocity and $\tilde{\omega} \sim \mathcal{N}(\omega, \sigma_\omega^2)$ for angular velocity, where $v$ and $\omega$ are ground-truth in the simulator. Based on the empirical study [15], depth cameras suffer from precision and lateral noise, as well as invalid values at full sensor resolution (nan ratio). We model precision noise as $\mathcal{N}(0, p_0 + p_1 d + p_2 d^2)$ and lateral noise as $\mathcal{N}(0, l_0 + l_1 \cdot \theta/(\pi/2 - \theta))$, where $d$ is the measured depth of pixel $(u, v)$ and $\theta$ is its azimuth. The nan ratio is modeled by a uniform distribution $\mathcal{U}(0, \delta_{\mathrm{nan}})$.

## C   Sim-to-Deploy Experiments

During training, the IsaacGym simulator runs at 200 Hz, and the policy runs at 50 Hz. In simulation experiments, ROS Gazebo provides odometry at 1000 Hz. In real-world experiments with Jackal, Faster-LIO [13] with IMU preintegration [14] fuses Lidar and IMU for 200 Hz odometry. Both settings use a depth camera RealSense D435i with $[640 \times 480]$ resolution at 30 fps, synchronized to odometry using ROS approximate time.

### C.1   Simulation Training Setups

**Uneven Terrain Datasets.** To generate datasets, we utilize a digital elevation model (DEM) represented as a raster obtained from a real-world high-resolution topography dataset [2]. The entire map is seamlessly divided into tiles of size $[128 \times 128]$ with resolution $0.1$ m. 3000 training data for DDPM and 100 evaluation data for algorithmic performance are randomly selected within a specific geographical range, defined by latitude and longitude intervals: $[33.5874, -116.0058]$, $[33.5874, -115.9991]$, $[33.5929, -116.0058]$, $[33.5929, -115.9991]$. For sim-to-sim deployment experiment in ROS Gazebo, we get 30 terrains, each of size $[320 \times 320]$ with resolu-

---

[2]https://opentopography.org

tion $0.1\,\mathrm{m}$, from latitude and longitude intervals: $[44.1859, -113.8802]$, $[44.1881, -113.8803]$, $[44.1881, -113.8773]$, $[44.1859, -113.8773]$.

**Policy Parameterization.** Our privileged (teacher) and deployment (student) policies use an encoder-decoder architecture. The encoder utilizes the first 16 layers of ResNet-18 [9] to extract feature representations from the terrain elevation or depth image. The deployment policy employs a neural circuit policy (NCP) [10], specifically the closed-form continuous-time (CfC) network, to generate linear and angular commands. While Vanilla RNN, LSTM, GRU, and NCP achieve similar accuracy with sufficient learning steps, we choose NCP for its superior performance with significantly fewer parameters [10]. We show details of PPO in Table 3, networks in Fig. 6, and domain randomization in Table 4. This setting keeps consistent for sim-to-sim and sim-to-real experiments.

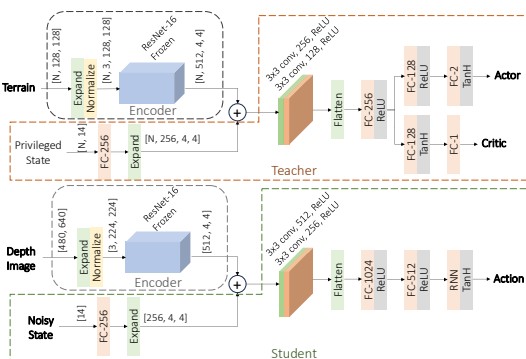

Figure 6: The network architecture for the teacher (privileged) and student (deployment) policies, with $N$ parallel robots training. The encoder uses the first 16 layers of ResNet-18, referred to as ResNet-16.

| Hyperparameter | Value | Hyperparameter | Value |
|---|---|---|---|
| Hardware Configuration | One RTX 4090 | PPO Clipping Parameter | 0.2 |
| Action Coefficient | [2.0, 1.4] | Optimizer | Adam |
| Discount Factor | 0.99 | Learning Rate | $5 \times 10^{-4}$ |
| Learning Epoch | 2 | Max Iterations | $1 \times 10^5$ |
| Generalized Advantage Estimation | 0.95 | Batch Size | $5 \times 10^5$ |
| Entropy Regularization Coefficient | 0.005 | Minibatch Size | $5 \times 10^4$ |

Table 3: Hyperparameters for Privileged Policy PPO.

| Parameter | Type | Distribution | Curriculum Range |
|---|---|---|---|
| **Environment** | | | |
| Dynamic Friction | Set | Gaussian | [0.1, 1.0] |
| Restitution | Set | Gaussian | [0, 0.2] |
| Gravity | Add | Gaussian | [0, 0.5] |
| **Robot** | | | |
| Mass | Add | Uniform | [0, 1.5] |
| Position | Add | Gaussian | [0, 0.05 |
| Orientation | Add | Gaussian | [0, 0.01] |
| Angular Velocity | Add | Gaussian | [0, 0.1] |
| Linear Velocity | Add | Gaussian | [0, 0.1] |
| **Robot-Environment Interaction** | | | |
| Depth Precision | Add | Gaussian | [0.0015, 0.015] |
| Depth Noise | Add | Gaussian | [0, 0.01] |
| Depth Nan Ratio | Set | Uniform | [0, 0.3] |
| External Force | Add | Gaussian | [0, 0.5] |
| Actuator | Add | Uniform | [0, 0.05] |
| **Procedural Environment Generation** | | | |
| Random Uniform | Set | Uniform | Height $[-0.45, 0.45]$, Step $[0.005, 0.045]$ |
| Slope | Set | Uniform | $[0.05, \sqrt{3}/2]$ |
| Discrete Obstacles | Set | Uniform | Height $[0.4, 10]$, Size $[0.1, 2.0]$, Num. $[1, 20]$ |
| Wave | Set | Uniform | Num. $[1, 20]$, Amp. $[0.1, 4/\mathrm{Num}]$ |

Table 4: Domain Randomization Parameters.

Additionally, we justify the procedural environment generation implemented in [11], where the height data type should be changed from integer to double to accommodate our parameter ranges. In the Random Uniform terrain, "step" represents the maximum height difference between two adjacent elevation grids. We choose $0.045\,\mathrm{m}$ as the upper step limit because it is consistent with Jackal's dimensions, where the chassis-to-ground distance is $0.058\,\mathrm{m}$ based on our measurements. Wave

environment generates terrains based on the trigonometric functions. Similarly, other procedurally generated environments are also navigable.

## C.2 Sim-to-Sim Experiment

In addition to benchmarking on the ClearPath Jackal robot, we train the ClearPath Husky robot using the same settings as the Jackal and tested in the same ROS Gazebo environments E-30 detailed in Sec. C.1. We evaluated across 30 environments, each containing 1000 pre-sampled start-goal pairs, resulting in total 30 000 trials. The results in Table 5 demonstrate that our method generalizes effectively across different wheeled platforms, maintaining an advantage over other methods in terms of success rate, orientation vibration, and position jerk. All methods performed better overall with the Husky, due to its better navigability on uneven terrains than Jackal. However, the performance differences between our method, Natural Adaptive Terrain (N-AT), and Procedural Generation Curriculum (PGC) relative to other methods mirror those observed in the Jackal experiment, with challenges such as perception noise and erratic prediction persisting for competing methods. In summary, our method generalizes well across different wheeled platforms and maintains a performance advantage over baseline methods.

| | Succ. Rate (%) | Traj. Ratio | Orien. Vib. ($\frac{rad}{s}$) | Orien. Jerk ($\frac{rad}{s^3}$) | Pos. Jerk ($\frac{m}{s^3}$) |
|---|---|---|---|---|---|
| Falco [16] | 53 | 1.47 | 1.85 | **91.6** | 33.6 |
| MPPI [17] | 65 | **1.32** | 0.74 | 145.5 | 33.7 |
| TERP [18] | 46 | 1.54 | **0.7** | 141.1 | **33.1** |
| POVN [19] | 43 | 2.5 | 1.28 | **121.4** | 34.6 |
| N-AT | **69** | **1.31** | 0.98 | 170.3 | 42.1 |
| PGC | 68 | 2.1 | 1.05 | 233.7 | 42 |
| Ours | **85** | 1.34 | **0.65** | 125.7 | **32** |

Table 5: Statistical results for Husky Gazebo simulation comparing our method with baselines of PGC and N-AT as well as previous works. 30 environments have 1000 start-goal pairs in each. **Green** and **Bold** indicate the best and second-best results.

## C.3 Sim-to-Real Experiment

**Wheeled Robot Navigation.** In the Jackal real-world experiment, we ablate the physics (**w/o Physics**) and perception (**w/o Percept**) domain randomization (DR) to study their contributions, shown in Table 6 and Table 7. We observed varying contributions of physics and perception domain randomization. First, in `forest`, `mud`, and `gravel`, removing either domain decreased the success rate, but these ablations still performed better than baselines due to ADTG. In `arid` and `rust`, the perception domain was crucial because of increased perception noise. Second, we identified drawbacks in the perception domain, particularly in `dunes` where the camera depth quality heavily depended on exposure, which is difficult to model. Last, without the physics domain, the robot showed improved orientation smoothness but struggled to make reactive behaviors, since the learned arc movements in slippery areas benefited from this domain. Additionally, for computational efficiency per frame, our method averaged $28.68\,\mathrm{ms}$ over $10^4$ frames, compared to MPPI's $20.78\,\mathrm{ms}$, Falco's $50.15\,\mathrm{ms}$, and POVNav's $157.22\,\mathrm{ms}$.

| | Suc. Rate (%) | | Traj. Ratio | | Orien. Vib. ($\frac{rad}{s}$) | | Orien. Jerk ($\frac{rad}{s^3}$) | | Pos. Jerk ($\frac{m}{s^3}$) | |
|---|---|---|---|---|---|---|---|---|---|---|
| | R | D | R | D | R | D | R | D | R | D |
| Ours | **80** | **45** | 1.25 | 1.32 | **0.11** | **0.13** | 113.3 | **73.4** | **20** | 18.1 |
| Ours w/o Physics DR | **74** | 42 | **1.18** | 1.35 | **0.1** | 0.13 | **112.6** | 74.2 | 25 | 20.3 |
| Ours w/o Percept DR | 72 | **48** | 1.27 | 1.31 | 0.11 | **0.12** | 117 | **72.5** | 24.6 | **17.9** |

Table 6: Statistical results for R(eal-world) and D(une) comparing our method with ablations of physics and perception domain randomization. R: grass, forest, arid, rust, mud, and gravel; D: dune hard, tough, and expert. Each environment for each method has 36 start-goal pairs. **Green** and **Bold** indicate the best and second-best.

The issue of imprecise depth measurements is highlighted in Fig. 7, where portions of the elevation map appear above the trajectories due to depth measurements inaccurately portraying unstructured objects. Compared to baselines that rely on elevation maps, our method showed more success. This is because the perception randomization during training introduces depth noises, allowing the

| | Success Rate (%) | | | | | | | | |
|---|---|---|---|---|---|---|---|---|---|
| | Grass | Forest | Arid | Rust | Mud | Gravel | Dune-Hard | Dune-Tough | Dune-Expert |
| Falco [16] | 100 | 25 | 58 | 33 | 39 | 28 | 31 | 17 | 17 |
| MPPI [17] | 100 | 22 | 92 | **56** | **86** | 69 | **75** | 17 | 11 |
| TERP [18] | 100 | 19 | 86 | 53 | 83 | 64 | 53 | 11 | 11 |
| POVN [19] | 100 | 0 | 14 | 22 | 17 | 14 | 42 | 14 | 17 |
| Ours | 100 | **33** | **100** | **58** | **97** | **92** | **75** | **39** | **22** |
| Ours w/o Physics DR | 100 | **28** | **97** | **58** | 78 | 81 | **72** | **36** | 17 |
| Ours w/o Percept DR | 100 | 25 | 89 | 50 | 81 | **89** | **72** | **39** | **33** |

Table 7: Detailed results of success rate among Falco, MPPI, TERP, POVNav, and our method with ablations of physics and perception domain randomization. Each environment for each method has 36 start-goal pairs.

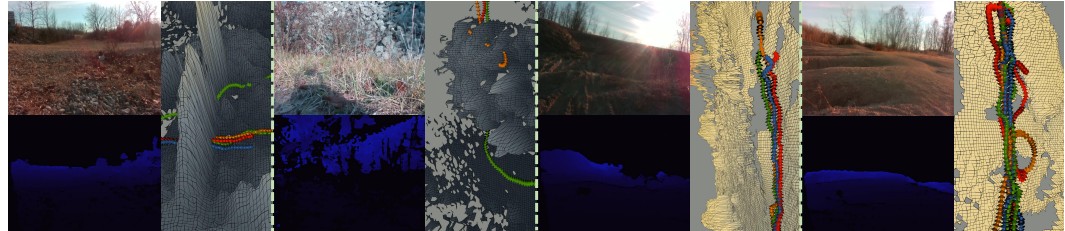

Figure 7: Qualitative presentation of four real-world experiments, with each featuring rectified RGB and depth images, accompanied by an elevation map marked with four distinct trajectories: [Green] Ours, [Orange] Falco, [Red] MPPI, [Blue] TERP. From left to right: [Arid] [Mud] Characterized by dense shrubbery, this scenario highlights our method's ability to navigate slippery conditions and obstacles effectively. [Dune-Tough] [Dune-Expert] Our method's preference for flatter regions proves to be a safer strategy in an environment fraught with unpredictable ravines.

policy to be more robust against misleading depth measurements. Similar results are also observed in a recent work for legged locomotion [5]. The complex vehicle-terrain interactions posed another challenge. As shown in Fig. 7, our method's trajectories had more curves, representing reactive behavior in slippery areas and leading to the relatively large trajectory ratio in Table 1. The learned arc movements are simple but effective, which will be explored further in our future research.

**Quadruped Robot Locomotion.** Refer to Parkour [12] Appendix B for detailed training in simulation. The parameters that differ from Parkour are as follows, with all other settings remaining the same. We set the maximum iteration of PPO to $1 \times 10^5$ and the number of parallel environments to 200. For the environment, the terrain size is $[128 \times 128]$ with a horizontal scale of $0.1$. The nine experimental environments with various challenges, including slippery surfaces, sloped terrains, and unpredictable ravines. These environments are the same as those used in the ground vehicle experiment but were traversed continuously in a loop. The gaits of our ADTG and the built-in Go1 MPC were natural, unlike the Procedural Generation Curriculum (PGC), which showed robust but prostrate postures. The PGC policy performed well but faced limitations due to sudden movements (jumps) that caused the robot to roll over, as shown in Fig. 8. PGC's issue may originate from its over-challenging unrealistic environments and we propose to investigate in the future.

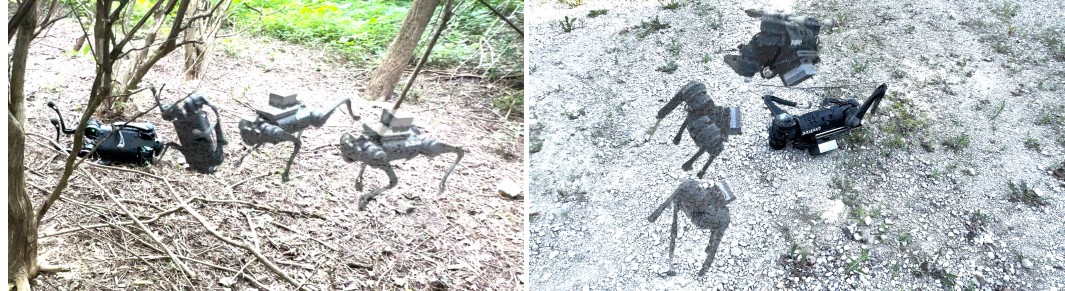

Figure 8: Failure case of the Procedural Generation Curriculum policy due to rapid movements.

# Appendix References

[A1] Kappen, Hilbert J., Vicenç Gómez, and Manfred Opper. "Optimal control as a graphical model inference problem." Machine learning 87 (2012): 159-182.

[A2] Levine, Sergey. "Reinforcement learning and control as probabilistic inference: Tutorial and review." arXiv preprint arXiv:1805.00909 (2018).

[A3] Uehara, Masatoshi, Yulai Zhao, Tommaso Biancalani, and Sergey Levine. "Understanding Reinforcement Learning-Based Fine-Tuning of Diffusion Models: A Tutorial and Review." arXiv preprint arXiv:2407.13734 (2024).

[A4] Ben-Hamu, Heli, Omri Puny, Itai Gat, Brian Karrer, Uriel Singer, and Yaron Lipman. "D-Flow: Differentiating through Flows for Controlled Generation." arXiv preprint arXiv:2402.14017 (2024).

[A5] Ho, Jonathan, Ajay Jain, and Pieter Abbeel. "Denoising diffusion probabilistic models." Advances in neural information processing systems 33 (2020): 6840-6851.

[A6] Nichol, Alexander Quinn, and Prafulla Dhariwal. "Improved denoising diffusion probabilistic models." In International conference on machine learning, pp. 8162-8171. PMLR, 2021.

[A7] Schulman, J., Wolski, F., Dhariwal, P., Radford, A., and Klimov, O. (2017). Proximal policy optimization algorithms. arXiv preprint arXiv:1707.06347.

[A8] Ross, S., Gordon, G., and Bagnell, D. (2011, June). A reduction of imitation learning and structured prediction to no-regret online learning. In Proceedings of the fourteenth international conference on artificial intelligence and statistics (pp. 627-635). JMLR Workshop and Conference Proceedings.

[A9] He, K., Zhang, X., Ren, S., and Sun, J. (2016). Deep residual learning for image recognition. In Proceedings of the IEEE conference on computer vision and pattern recognition (pp. 770-778).

[A10] Lechner, M., Hasani, R., Amini, A., Henzinger, T. A., Rus, D., and Grosu, R. (2020). Neural circuit policies enabling auditable autonomy. Nature Machine Intelligence, 2(10), 642-652.

[A11] Makoviychuk, Viktor, Lukasz Wawrzyniak, Yunrong Guo, Michelle Lu, Kier Storey, Miles Macklin, David Hoeller et al. "Isaac gym: High performance gpu-based physics simulation for robot learning." arXiv preprint arXiv:2108.10470 (2021).

[A12] Zhuang, Ziwen, Zipeng Fu, Jianren Wang, Christopher Atkeson, Soeren Schwertfeger, Chelsea Finn, and Hang Zhao. "Robot parkour learning." arXiv preprint arXiv:2309.05665 (2023).

[A13] Bai, C., Xiao, T., Chen, Y., Wang, H., Zhang, F., and Gao, X. (2022). Faster-LIO: Lightweight tightly coupled LiDAR-inertial odometry using parallel sparse incremental voxels. IEEE Robotics and Automation Letters, 7(2), 4861-4868.

[A14] Y. Yu et al., "Fast Extrinsic Calibration for Multiple Inertial Measurement Units in Visual-Inertial System," 2023 IEEE International Conference on Robotics and Automation (ICRA), London, United Kingdom, 2023, pp. 01-07.

[A15] G. Halmetschlager-Funek, M. Suchi, M. Kampel and M. Vincze, "An Empirical Evaluation of Ten Depth Cameras: Bias, Precision, Lateral Noise, Different Lighting Conditions and Materials, and Multiple Sensor Setups in Indoor Environments," in IEEE Robotics & Automation Magazine, vol. 26, no. 1, pp. 67-77, March 2019.

[A16]  Zhang, Ji, Chen Hu, Rushat Gupta Chadha, and Sanjiv Singh. "Falco: Fast likelihood-based collision avoidance with extension to human-guided navigation." Journal of Field Robotics 37, no. 8 (2020): 1300-1313.

[A17]  Williams, Grady, Paul Drews, Brian Goldfain, James M. Rehg, and Evangelos A. Theodorou. "Aggressive driving with model predictive path integral control." In 2016 IEEE International Conference on Robotics and Automation (ICRA), pp. 1433-1440. IEEE, 2016.

[A18]  Weerakoon, Kasun, Adarsh Jagan Sathyamoorthy, Utsav Patel, and Dinesh Manocha. "Terp: Reliable planning in uneven outdoor environments using deep reinforcement learning." In 2022 International Conference on Robotics and Automation (ICRA), pp. 9447-9453. IEEE, 2022.

[A19]  Pushp, Durgakant, Zheng Chen, Chaomin Luo, Jason M. Gregory, and Lantao Liu. "POV-Nav: A Pareto-Optimal Mapless Visual Navigator." arXiv preprint arXiv:2310.14065 (2023).

