# OpenReview forum: "Adaptive Diffusion Terrain Generator for Autonomous Uneven Terrain Navigation"
_robot-learning.org/CoRL/2024/Conference — CoRL 2024_

### Official Review · Reviewer_NMmZ · 2024-07-21
**Review of Submission166**

**Originality:** 3
**Technical Quality:** 1
**Clarity Of Presentation:** 3
**Potential Impact:** 3
**Recommendation:** 2
**Confidence:** 5

**Review:**

The writing is clear and the paper provides insights into terrain generation in software environment via reinforcement learning.
The authors claim that their terrain generation approach can capture the contact dynamics of very complex terrains and they try to show their method's applicability via simulations and robot experiments. Although the context is original and highly relevant to the robotics community, as it attempts to address the significant challenge of bridging the sim-to-real gap, I have concerns regarding the feasibility and scientific rigor of their methods. Specifically, capturing the terra-dynamics of complex, deformable, and flowable surfaces (such as sand, mud, and snow) is a very challenging task due to the highly nonlinear interactions of millions of particles during robotic limb contact. Also such interactions are bound to hysteresis effects, i.e., the terra-dynamics are different if the robot encounters a previously disturbed terrain. Researchers often attempt to capture these interactions using experimental methods, which only partially work and rely on various assumptions. See the details of such an attempt in the paper below:

Li et al, A terradynamics of legged locomotion on granular media, 2013

Although the proposed approach claims to outperform other methods, it is unclear whether this performance increase is due to the proposed control policy itself or other parameters related to robot-terrain interactions. And such a task would require extensive investigations of granular physics which I believe is out of the scope for CORL.

Strengths:
Model-free reinforcement learning is highly effective for developing robust robot control policies for navigating complex terrains.

The use of massively parallel physics simulations speeds up policy training significantly.

Weaknesses:
There are concerns regarding the feasibility and scientific rigor of the proposed methods.

Capturing the terra-dynamics of complex, deformable, and flowable surfaces (such as sand, mud, and snow) is highly challenging, and current methods only partially work under various assumptions.

The evaluation of the robots' performance is not clearly presented.

**Quality Of The Limitations Section:**

1

**Questions For Rebuttal:**

Fig 1, fig 4 and fig 6 are small hard to grasp.

It is unclear how section 4 generates flowable and deformable surfaces such as sand and mud.

Section 5.2:
The metrics are not clearly explained.
The details regarding the robot's parkours in both sim and exp subsections are not given. Also, I realized that the mud terrain does not seem to be wet in the video.
Limitations are not addressed.
The conclusion section does not seem to reflect the key messages of the paper.

**Robotics Focus:**

4

**Summary Of Paper:**

This paper presents a model-free reinforcement learning approach proposing robust robot control policies for navigating complex terrains, relying on parallel physics simulations and environment generators. The Adaptive Diffusion Terrain Generator (ADTG) presented here addresses the limitations of traditional approaches of parameter-based heuristics by using Denoising Diffusion Probabilistic Models (DDPMs) to create diverse, complex terrains tailored to the policy's abilities. ADTG blends terrains with performance-informed weights and manipulates initial noise to generate suitable challenges and novel environments. Experimental results claim that ADTG-trained policies outperform those trained with traditional methods and natural environments.

**Summary Of Recommendation:**

I have concerns regarding the feasibility and scientific rigor of their methods, particularly in capturing the terra-dynamics of complex, deformable, and flowable surfaces like sand and mud. These interactions are nonlinear and influenced by hysteresis effects, making accurate predictions challenging. While the authors claim their method outperforms existing ones, it is unclear if this is due to the proposed control policy or other factors related to robot-terrain interactions. This area requires extensive investigations of granular physics, which may be beyond the scope of this work.

---

### Official Review · Reviewer_RP3g · 2024-07-30
**Quality paper advancing wheeled robots' ability to traverse rough terrain.**

**Originality:** 3
**Technical Quality:** 4
**Clarity Of Presentation:** 5
**Potential Impact:** 3
**Recommendation:** 3
**Confidence:** 4

**Review:**

The paper is presented well and covers most important points to justify that the proposed method is an improvement on SOTA.

Strengths:

- combines diversity and difficulty adjustment intuitively into the diffusion framework

- the task of uneven terrain nav w/ wheeled robots is an important area for robotics

- Their results show a substantial improvement (10-20% success rate improvements)

- Quality of implementation is good and figures are very illustrative

- Tested on a real platform in a variety of environments (e.g. grass, forest, gravel, sand)

- Blending approach is verified as an approximation of terrain difficulty in appendix

Weaknesses:

- Only one wheeled platform is tested (Jackal). There may be some variability with other wheeled configurations

- A quadruped (Unitree Go1) is tested, while the author's method seems to perform better, the results are very limited (no tables presented, only mentions number of failures but some experimental details are missing (e.g. the time the robots were tested in each experiment)

- The main baseline of this paper is the Procedural Content Generation (PCG) (reference 13) which was intended for quadrupeds. The terrain generation parameters may not be appropriate for a wheeled robot, e.g. the author includes steps of heights 0.1-0.3m, which may be impossible for this robot and hinder the performance of the baseline due to the discrepancy in robot morphology

- No limitations are discussed explicitly in this paper

**Quality Of The Limitations Section:**

1

**Questions For Rebuttal:**

1. Are the PCG parameters and terrain types appropriate for the Jackal? Are you able to verify this or ablate them? If the ADTG policy can't traverse a reasonable range of these PCG terrain parameters then these should be adjusted for fair comparison.

2. Are there limitations of your proposed method?

3. Were the quadruped experiments ran in the same was as the Jackal experiments (sampling 36 start-goal pairs per experiment)?

**Robotics Focus:**

4

**Summary Of Paper:**

This paper presents a diffusion-based terrain generation and assignment algorithm. The author's theorise that by providing a terrain generator that is both realistic and flexible will improve policy performance when trained with reinforcement learning.  They ablate their methods with existing work that uses parameterised terrains (e.g. steps, perlin noise), and show that non-parametric diffusion generators outperform the parameterised methods. They further ablate their methods with existing navigation methods, and do extensive real world training. They propose a blending strategy to blend the latent space of two (or more?) terrain embeddings to get new terrains with varied traverse difficulties. They also propose adjusting the diffusion denoising steps to gradually increase diversity over time.

**Summary Of Recommendation:**

If the authors provide a justification for the baseline (PCG) parameters used or an ablation to show these are not unfairly disadvantaging the baseline, I would strongly recommend this paper for CoRL. Without this justification, I am uncertain but leaning towards 'accept' as many non-curriculum based baselines are compared and the contributions are valuable nonetheless.

---

### Official Review · Reviewer_1ULr · 2024-07-30

**Originality:** 5
**Technical Quality:** 4
**Clarity Of Presentation:** 5
**Potential Impact:** 4
**Recommendation:** 4
**Confidence:** 4

**Review:**

This paper is well written and ideas are presented clearly. The results are thorough, with significant ablations and comparisons. Extensive experiments across simulations, and diverse terrain types and robot embodiment's in challenging real world scenarios validate the robustness of this method compared to several baselines. The work from this paper offers an exciting terrain generation method and curriculum-based learning approach that will clearly benefit the robot learning community.

I couldn't see an explicit report of limitations, I'm not sure if this is still a requirement for this conference.

Minor feedback:

In Section 4.1 and 4.2, ‘e’ is used for the DPPM latent variable and for individual terrain maps. Are they the same size, i.e a latent variable for each pixel of a terrain map and corresponding weighting term? Adding some clarity will help distinguish between these variables.

What is “trajectory ratio” in Table 1?

In Algorithm 1, is the “Optim” step for a single rollout, or to convergence for a given terrain map? How many terrains are trained in parallel, and does each robot have a different terrain instance?

For each Policy Update a new terrain is added, the number of terrains in Lambda would quickly grow, is the variance calculation of 4.2 performed over the entire set of terrains? How long does it take to run through each stage in Algorithm 1, what's the most compute/time intensive step, and the overall training time?

“we propose the follows:” -> “we propose the following:”

**Quality Of The Limitations Section:**

2

**Questions For Rebuttal:**

See above

**Robotics Focus:**

4

**Summary Of Paper:**

This paper introduces Adaptive Diffusion Terrain Generator (ADTG), a two-stage approach to generating terrain maps for training policies in simulation (IsaacGym) for effective learning and robust deployment to other simulation (Gazebo), and real world. The first stage is to train a Denoising Diffusion Probabilistic Model from 3000 representative terrain maps based on real terrain data, and using this model to blend terrains from the initial dataset with generated maps with a weighting determined from policy success. The second is to add more diverse and complex terrains by manipulating the noise in the diffusion process.

**Summary Of Recommendation:**

Clearly a standout work with novel contributions and significant experimental results on hardware.

---

### Official Review · Reviewer_rh2f · 2024-08-01
**Terrain Generation could be a potential good topic for robustness, the algorithm design lacks further justification.**

**Originality:** 4
**Technical Quality:** 3
**Clarity Of Presentation:** 3
**Potential Impact:** 3
**Recommendation:** 2
**Confidence:** 3

**Review:**

The key contributions of this work include:
 ADTG synthesizes terrains of varying difficulties by blending terrains from an initial dataset within their latent spaces, guided by performance-informed weights.
 By manipulating the initial noise in the diffusion process, ADTG shifts between creating similar terrains for fine-tuning and entirely novel ones for expanding training diversity.
 Experiments demonstrate that policies trained with ADTG outperform those trained in procedural or natural environments, showing enhanced generalization capabilities and faster convergence.
 ADTG integrates with adaptive curriculum reinforcement learning (ACRL), co-evolving the terrain dataset and the robot's policy to continuously challenge and improve the robot's capabilities.
The proposed method addresses limitations in existing terrain generation techniques by providing a more diverse and realistic training environment, thereby enhancing the robustness and adaptability of autonomous navigation policies.

**Quality Of The Limitations Section:**

2

**Questions For Rebuttal:**

1. Eq.1, the definition is not clear. The expectation is over what?(Policy or terrain parameters?)
2. A deeper explanation of the design choices in ADTG, such as the selection of DDPMs over other generative models, and the specific weighting function used for blending terrains, would provide a better understanding of the underlying principles and their impact on performance.
3. While the experiments demonstrate the effectiveness of ADTG, additional real-world validation across a broader range of scenarios and environments could further substantiate the claims.

**Robotics Focus:**

4

**Summary Of Paper:**

The paper introduces the Adaptive Diffusion Terrain Generator (ADTG), a novel environment generation method for training autonomous robots to navigate uneven terrains. This method utilizes Denoising Diffusion Probabilistic Models (DDPMs) to create diverse and challenging terrains tailored to the robot's current capabilities. Unlike conventional heuristic-based methods, ADTG dynamically adjusts terrain complexity and variety based on the robot's evolving performance.

**Summary Of Recommendation:**

While the Adaptive Diffusion Terrain Generator (ADTG) presents an innovative approach to terrain generation, there are areas where the justification of the algorithm design could be strengthened.

---

### Author Rebuttal · Authors · 2024-08-10

Dear Reviewers and Area Chair:\
Thank you very much for reviewing our paper, and thank you for your patience.  We sincerely appreciate the constructive feedback provided by all reviewers, which will undoubtedly enhance the quality of our work. In this response file, we address each comment made by the reviewers point by point.  Modifications in the manuscript are highlighted in red color so they can be easily located and matched to the revised version. We have finished the complete result of the Husky simulation experiment, which includes 30 environments with 30000 start-goal pairs in total. We have accordingly added the details of settings, results, and evaluations in the manuscript.

---

### Decision · Program_Chairs · 2024-09-04

**Decision:**

Accept

**Comment:**

This paper has been reasonably well-received by the reviewers.  The paper is relatively clear and well-motivated.

The reviewers appreciated the diversity of different platforms this was tested on (with the exception of reviewer 2 pointing out a single wheeled platform - more wheeled platforms would be appreciated).

The comparison to other navigation algorithms, and the use of real robots for testing are other strong points.

Experimentation is generally good, with a number of different studies undertaken.

I have some reservations about how scalable this is - in that the terrain maps used are presumably of a given feature size, and thus would only produce maps of a similar feature size.  Can the authors please comment?

The testing section is weak.
-There is a lack of specifics of the testing and evaluation - e.g., what is a failure? Some of the experimental and testing settings are missing (see reviewer 2)

-There is a lack of forthright discussion on limitations

I encourage the authors to fully address the reviewer comments to increase chances of acceptance.  Specifically, addressing limitations, adding more detail to the experimental session, and showing more relevant information will help to increase scientific rigor.


Update:

The authors have responded well and improved their submission in response to reviewer comments.